# Comparison of Three Glycoproteomic Methods for the Analysis of the Secretome of CHO Cells Treated with 1,3,4-*O*-Bu_3_ManNAc

**DOI:** 10.3390/bioengineering7040144

**Published:** 2020-11-10

**Authors:** Joseph L. Mertz, Shisheng Sun, Bojiao Yin, Yingwei Hu, Rahul Bhattacharya, Michael J. Bettenbaugh, Kevin J. Yarema, Hui Zhang

**Affiliations:** 1Wilmer Eye Institute, Johns Hopkins University, Baltimore, MD 21287, USA; jmertz2@jhmi.edu; 2Department of Pathology, Johns Hopkins University, Baltimore, MD 21287, USA; shisheng_sun@126.com (S.S.); yhu39@jhmi.edu (Y.H.); 3Department of Chemical and Biomolecular Engineering, Johns Hopkins University, Baltimore, MD 21218, USA; yinbojiao@gmail.com (B.Y.); beten@jhu.edu (M.J.B.); 4Department of Biomedical Engineering, Johns Hopkins University, Baltimore, MD 21218, USA; rahul.chemiit@gmail.com (R.B.); kyarema1@jhu.edu (K.J.Y.)

**Keywords:** glycoproteomics, CHO, sialylation, 1,3,4-*O*-Bu_3_ManNAc

## Abstract

Comprehensive analysis of the glycoproteome is critical due to the importance of glycosylation to many aspects of protein function. The tremendous complexity of this post-translational modification, however, makes it difficult to adequately characterize the glycoproteome using any single method. To overcome this pitfall, in this report we compared three glycoproteomic analysis methods; first the recently developed N-linked glycans and glycosite-containing peptides (NGAG) chemoenzymatic method, second, solid-phase extraction of N-linked glycoproteins (SPEG), and third, hydrophilic interaction liquid chromatography (HILIC) by characterizing N-linked glycosites in the secretome of Chinese hamster ovary (CHO) cells. Interestingly, the glycosites identified by SPEG and HILIC overlapped considerably whereas NGAG identified many glycosites not observed in the other two methods. Further, utilizing enhanced intact glycopeptide identification afforded by the NGAG workflow, we found that the sugar analog 1,3,4-*O*-Bu_3_ManNAc, a “high flux” metabolic precursor for sialic acid biosynthesis, increased sialylation of secreted proteins including recombinant human erythropoietin (rhEPO).

## 1. Introduction

Glycosylation is a widespread protein modification that plays a role in protein folding, stability, protein−ligand interactions, biological function, and pathogenesis [1,2,3,4]. Because each glycoprotein can exhibit multiple glycosylation sites (glycosites) and each glycosite can exhibit dozens or even hundreds of glycan structures, glycosylation is the most complex modification, with the potential for a single protein isoform to exhibit thousands of glycosylation combinations [5]. Further, specific glycan structures at specific glycosites have been associated with altered function and disease [6]. Thus, comprehensive systematic analysis is both challenging and crucial.

Virtually all therapeutic proteins are glycosylated—with notable exceptions such as insulin and human growth hormone—and glycosylation can influence the conformation, stability, pharmacokinetic profile, in vivo activity, and immunogenicity of recombinant proteins [7]. One significant and well-documented contribution of glycosylation to the pharmacokinetic profile—generally encompassing absorption, distribution, metabolism and excretion (ADME)—is increased half-life in circulation, which greatly affects dosing strategies and thus the ultimate success of a drug [8,9]. Terminal sialic acid residues, in particular, have been shown to increase circulatory half-life by masking galactose residues from galactose-specific receptors in the liver [10] and other would-be terminal glycan residues from lectin-like receptors elsewhere in the body [11,12].

Mammalian cell expression systems dominate therapeutic protein production with Chinese hamster ovary (CHO) cells by far the most prevalent, producing upwards of 70% of recombinant protein therapeutics [13]. This is due to their capacity for high-density culture in serum-free conditions, accumulation of biomanufacturing expertise surrounding these cells, the development of efficient gene introduction techniques, and glycosylation profiles that are compatible for use in humans [14]. Although the glycosylation machinery of CHO cells is similar enough to produce glycosylation patterns tolerated in human therapeutics, important differences do exist. One notable example is the lack of α-2,6-sialyltransferase (encoded in humans by *ST6GAL1*), which not only decreases overall sialylation but also deprives therapeutic proteins of α-2,6-sialylation which can negatively affect the pharmacodynamics of protein drugs in humans [15,16,17]. Sialylation in CHO cells is also limited by the relatively low proportion of *N*-glycan branches, which provide acceptor sites for sialylation, caused by low expression of *N*-acetylglucosaminyltransferases such as Mgat4 and Mgat5 [18]. Thus, strategies have been introduced to glycoengineer CHO cell culture systems to optimize glycosylation patterns both genetically and metabolically. Genetic approaches include exogenous expression of glycosylation machinery genes ST6GAL1 [19] to increase and humanize sialylation or Mgat4 and Mgat5 to increase *N*-glycan branching [20] while metabolic approaches involve additives to culture media to help drive desired forms of glycosylation [21].

Culture supplementation strategies designed to improve the glycosylation profile therapeutic proteins include controlling the mannose to glucose ratio [22], adding nucleotide sugar glycosylation precursors such as uridine diphosphate-N-acetylglucosamine (UDP-GlcNAc) and cytidine monophosphate-sialic acid (CMP-SA) [23], and supplying co-factors for glycosylating enzymes such as manganese [24]. Supplementing the culture media with the sialic acid precursor *N*-acetylmannosamine increases intracellular CMP-SA and improves the sialylation of interferon-gamma glycan structures [25]. Meanwhile, sodium butyrate (NaBu) a culture additive that has been used to increase recombinant protein yield in CHO cells [26], also demonstrates adverse effects on cell viability and glycosylation [27]. Interestingly, pitfalls associated with NaBu have been overcome by combining this chemical with ManNAc in a single molecule; specifically, 1,3,4-*O*-Bu_3_ManNAc. This synthetic analogue consists of ManNAc appended with ester-linked butyrate to the hydroxyl groups found at the 1, 3, and 4 positions of this monosaccharide. By masking the hydrophilic moieties of both the sugar and butyrate moieties, membrane permeability and concomitant cell uptake are increased by two orders of magnitude or more. As a result, 1,3,4-*O*-Bu_3_ManNAc strongly increases metabolic flux of ManNAc into the sialic acid biosynthetic pathway and achieves the benefits of butyrate on recombinant protein yield without deleterious effects on cell viability or glycan quality [27,28]. To date, 1,3,4-*O*-Bu_3_ManNAc has been used to improve the sialylation of erythropoietic (EPO) [27] immunoglobin G (IgG) antibodies [28]. While the overall effect of this analog on the glycoproteome of CHO cells has not been examined previously, it has been shown to boost cancer cells [29,30].

Glycoproteomics research has progressed rapidly over the last decade, propelled by advances in sample preparation techniques, mass spectrometry (MS), and data processing. Innovations in affinity-based capture of glycans, glycopeptides, and glycoproteins by lectins, antibodies, or chemical affinity resins [31,32] have increased coverage of the glycoproteome [33,34,35]. Solid-phase extraction of *N*-linked glycoproteins (SPEG), which utilizes covalent bonds between hydrazide beads and oxidized glycan moieties of *N*-glycosylated peptides followed by the release by PNGase F was one of the first examples of such a workflow, and it has seen continued use since its introduction in 2003 [35,36,37]. Hydrophilic interaction liquid chromatography (HILIC), is a collective term for purification techniques that generally utilize solid-phase resins with neutral polar moieties to retain an aqueous layer, and thereby sequester glycopeptides via their hydrophilic glycan moieties from a less polar mobile phase [38]. Subsequent PNGase F treatment and LC-MS/MS allows proteome-wide glycosite identification as in SPEG workflows.

Characterization of glycoproteins has now advanced to include intact glycopeptide analysis that can simultaneously analyze glycosites and their attached glycans [36,39,40]. The large search space created by the great diversity of the glycome makes it difficult to accurately identify intact glycopeptides and has led to various strategies to restrict or guide the peptide sequence matching process. A particularly powerful method known as solid-phase extraction of *N*-linked glycans and glycosite-containing peptides (NGAG) [41] combines chemical and enzymatic modifications of peptides to allow parallel glycosite and glycan analysis across the proteome and which then subsequent intact glycopeptide analysis bioinformatically. Briefly, after trypsin digestion glycopeptides are guanidinated to block lysine ε-amino groups and then covalently bound to an aldehyde solid phase via the terminal primary amine. They are then treated with aniline to modify carboxyl groups—including, importantly, aspartic acid and sialic acid residues. Glycosylated asparagines are then deglycosylated and deamidated to aspartic acid by PNGase F treatment and released glycans analyzed by MALDI-TOF MS. The peptides containing aspartic acid residues newly formed by deglycosylation are selectively released by Asp-N and analyzed by LC-MS/MS to identify glycosites. Finally, these glycosites and glycans are used to improve peptide sequence matching (PSM) in parallel intact glycopeptide analyses. Data analysis in the NGAG workflow is performed by the GPQuest software package [42,43], which is integrated with the NGAG protocol to match intact glycopeptide LC-MS/MS spectra to glycan and peptide databases generated from the same samples.

To characterize the glycoproteome detection afforded by the NGAG pipeline, we performed a comparison of NGAG glycosite results in more conventional HILIC and SPEG methods in parallel analyses of the CHO-K1 cell secretome. Further, to investigate the potential of shifting the sialylation profile of therapeutic proteins to improve their efficacy, we treated CHO-K1 cells expressing recombinant human erythropoietin (rhEPO) with the ManNAc analog 1,3,4-*O*-Bu_3_ManNAc and analyzed intact glycopeptides in their secretome using NGAG. Our results showed that NGAG is comparatively orthogonal to either HILIC or SPEG and 1,3,4-*O*-Bu_3_ManNAc was effective for increasing sialylation throughout the secretome—including upon rhEPO expressed by these cells.

## 2. Methods

### 2.1. CHO Cell Culture and Protein Harvest

CHO-K1 cells expressing recombinant human EPO were generated and cultured as described previously [39]. Briefly, two biological replicates were cultured in suspension with CD-CHO media (Thermo Fisher Scientific; Rockford, IL, USA) and 5 mM glutamine (Thermo Fisher Scientific), seeded at 400,000 cells/mL and grown for 3 days with and without 333 µM 1,3,4-*O*-Bu_3_ManNAc treatment. 1,3,4-*O*-Bu_3_ManNAc was synthesized in house as described previously [44]. The secretome was concentrated and buffer exchanged by 10 KD filter (Sigma-Aldrich; Saint Louis, MO, USA) and the cells were pelleted and washed twice with PBS. Proteins were harvested from each sample (both secretome and cell proteome) with 10 mL 8 M urea in 1 M NH_4_HCO_3_ (Sigma-Aldrich) followed by two rounds of sonication with 40–60 cycles per round. Proteins were reduced by 10 mM tris(2-carboxyethyl)phosphine (Thermo Fisher Scientific), alkylated with 16.5 mM iodoacetamide (Sigma-Adrich), and digested with sequencing-grade trypsin (Promega; Madison, WI, USA) at a 1:50 enzyme:substrate ratio. Peptides were desalted using a C18 solid phase extraction (SPE) cartridge (Waters; Milford, MA, USA) and dried by SpeedVac. An equal mass of peptides, 600 µg, from each sample was used for all glycoproteome workflows to allow cross-comparison. This amount was chosen to ensure adequate deglycosylated peptides or glycopeptides for triplicate LC-MS/MS runs after each extraction protocol.

### 2.2. SPEG Enrichment of Deglycosylated Peptides

SPEG enrichment followed previous reports [35]. Briefly, tryptic peptides from the secretome were treated with 10 mM NaIO_4_ in 0.1% TFA for 1 h in the dark, purified by C18 column, followed by adding 1% aniline. Peptides were then incubated with 100 µL hydrazide beads (Bio-Rad; Hercules, CA, USA) per 1 mg peptides overnight at pH < 6. The beads were washed three times each with 50% ACN, 1.5 M NaCl, H_2_O and 25 mM NH_4_HCO_3_ with 20 s vortexing during each wash. Peptides were released from their glycans and the beads by 1000 units PNGase F (New England Biolabs; Beverly, MA, USA) cleavage overnight in 25 mM NH_4_HCO_3_. The supernatant plus one wash of the beads with H_2_O and one with 50% ACN was collected for analysis of glycosite analysis using mass spectrometry. Cell lysates were also analyzed by this method, following the same approach.

### 2.3. HILIC Enrichment of Deglycosylated Peptides and Intact Glycopeptides

Glycopeptides from each secretome sample were enriched using a HILIC column (SeQuant, Southborough, MA, USA) [45]. After tryptic peptides were dried, they were reconstituted in a final solvent composition of 80% ACN/0.1% trifluoroacetic acid (TFA, Sigma Aldrich). The HILIC column was conditioned twice with 0.1% TFA and twice with 80% ACN/0.1% TFA, the sample was loaded and washed three times with 80% ACN/0.1% TFA, and the bound glycopeptides then were eluted in 0.1% TFA. The samples were dried via SpeedVac and resuspended in 0.2% formic acid (FA, Sigma-Aldrich) if they were to be used for intact glycopeptide LC-MS/MS analysis or 25 mM NH_4_HCO_3_ for glycosite analysis. For glycosite analysis, *N*-glycans were cleaved by PNGase F digestion, 1000 units, overnight, then deglycosylated peptides were concentrated and purified by C18 column. The deglycosylated peptides were eluted from the C18 column, dried, and resuspended in 0.2% FA for LC-MS/MS. Glycosites were also analyzed from cell lysates by this method, following the same approach.

### 2.4. NGAG Enrichment of Deglycosylated Peptides and Glycans

Deglycosylated peptides and glycans were isolated from the secretome of each sample by NGAG, as described previously [41]. Lysines on tryptic peptides from each sample were guanidinated using 1.425 M NH_3_, and 0.6 M *O*-methylisourea (Sigma-Aldrich) for 30 min at 65 °C then purified by C18 column. Guanidinated peptides were conjugated by their amino terminals to 500 mL AminoLink beads (Pierce; Rockford, IL, USA) per 1 mg peptide overnight at RT. Carboxyl groups—on both glycans and the peptide backbones—were modified by aniline in the presence of 0.5 M 1-ethyl-3-(3-dimethylamin-opropyl)carbodiimide (EDC) in 200 mM MES buffer, pH 5, at RT overnight. To ensure that peptides with non-modified aspartate residues did not contaminate downstream steps, these were removed by Asp-N at 37 °C overnight, then *N*-glycans were released by overnight digestion with 1000 units PNGase F, and set aside for glycan analysis. Deglycosylated peptides were released at their nascent aspartate residues by overnight digestion with Asp-N (Promega) in 25 mM NH_4_HCO_3_ and then collected in the supernatant and prepared for glycosite analysis by LC-MS/MS.

### 2.5. Mass Spectrometry Analysis of Glycans

*N*-Glycans, after release by PNGase F within the NGAG workflow, were concentrated and purified using HyperSep Hypercarb SPE cartridges (Thermo Fisher Scientific). The columns were conditioned with 100% acetonitrile and 1% TFA, the sample was then loaded in 0.1% FA, washed with 0.1% FA, and eluted in 80% ACN/0.1% FA, then resuspended in deionized water. Samples of this resuspension were spotted onto a MALDI plate along with equal volume 2,5-dihydroxybenzoic acid/*N*,*N*-dimethylaniline (DHB/DMA, Sigma-Aldrich) matrix containing 100 mg/mL DHB, 2% DMA in 50% acetonitrile and 0.1 mM NaCl. Analyses were performed on an Axima MALDI Resonance mass spectrometer (Shimadzu, Columbia, MD, USA) with the laser power set to 100 for 200 total shots per spot. Glycan structures were elucidated from the most abundant peaks in the 800–2000 Da and 2000–3000 Da spectra by Glyco-Peakfinder (EuroCarbDB [46]).

### 2.6. LC-MS/MS Analysis

Deglycosylated peptides from SPEG and HILIC were analyzed on a Velos Pro mass spectrometer while intact glycopeptides and NGAG deglycosylated peptide samples were analyzed on a Q-Exactive mass spectrometer (Thermo Fisher Scientific), with 1 µg sample loading for all runs. For the Orbitrap Velos Pro analyses (SPEG and HILIC glycosite analyses), peptides were separated on an UltiMate UPLC system with a 75 μm × 15 cm Acclaim PepMap100 separating column (Thermo Scientific) at 300 nL/min. For the Q-Exactive analyses (NGAG glycosites and intact glycopeptides), peptides were separated on a Dionex Ultimate 3000 RSLC nano system with a 75 μm × 15 cm Acclaim PepMap100 separating column (Thermo Scientific) at 250–290 nL/min. Buffer A consisted of 0.1% FA and Buffer B 0.1% FA/95% acetonitrile.

For SPEG and HILIC glycosite experiments, separation was achieved by a gradient profile of 4–8% Buffer B over 6 min, 8–35% B over 84 min, 35−45% B over 10 min, 45–90% over 10 min, and 10 min at 90%. Electrospray voltage was 2.2 kV. MS1 spectra were collected from 400 to 1800 *m*/*z* with a resolution of 30,000. The top 10 most abundant ions with a 25 s dynamic exclusion were fragmented by HCD and MS2 were collected at 7500 resolution with 100 *m*/*z* fixed first mass.

For NGAG glycosite analysis, deglycosylated peptides were separated by a gradient profile of 4–10% Buffer B over 6 min, 10–28% over 84 min, 28–35% over 10 min, 35–90% over 10 min, 90% for 8 min, 90–4% over 2 min, then 10 min at 4%. The intact glycopeptide analysis utilized a slightly steeper gradient profile with 4% Buffer B for 5 min, 4–25% B over 95 min, 25–95% B over 5 min, 95% for 10 min, 95–45 over 1 min, followed by 4 min at 4%. Analysis by MS/MS of these samples was performed with 1.8–2.0 kV electrospray voltage. MS1 spectra were collected in the Orbitrap with AGC target of 3 × 10^6^ and 70,000 resolution from 400 to 2000 *m*/*z*. MS2 spectra were collected using HCD fragmentation of the top 9–15 most abundant ions at 17,500 resolution and 100 *m*/*z* fixed first mass. A dynamic exclusion of 15 s was used for intact analysis and 30 s for NGAG glycosite analysis.

### 2.7. Glycosite Data Analysis

Glycosite LC-MS/MS data were searched by MaxQuant (v1.5.8.3, Cox lab, Max-Planck Institute for Biochemistry, Martinsried, Germany [47]) against the Uniprot *Cricetulus griseus* proteome database downloaded 5 May 2016. SPEG and HILIC glycosite data were searched using up to two missed cleavages, 20 ppm peptide mass tolerance for the first search and 4.5 ppm for the main search, carbamidomethylation of C (+57.0215 Da) as a static modification, oxidation of M (+15.9949 Da) and acetylation of protein N-terminal (+42.0106 Da) as dynamic modifications, five modifications per peptide, a minimum peptide length of seven amino acids, and 1% FDR against a reversed decoy proteome database. Deamidation of N (+0.9840 Da) was added as a dynamic modification.

NGAG glycosite data searches used a custom proteome created by replacing the N of all potential *N*-glycosylation sites (N-X-S/T motifs, X ≠ P) with the dummy amino acid “U” of nominal mass of 168.9642 Da. This mass was chosen so that it will not match with commonly observed amino acids—with or without commonly observed modifications—in LC-MS/MS data and only matches with glycosite N residues that are deamidated to D at peptide N-termini (see dynamic modifications of U described below). Searches for NGAG also used a custom enzyme “Trypsin + U” created on the MaxQuant enzyme list, with cleavage at the N-terminal side of “U” in addition to trypsin K and R cleavage sites. The U amino acid created cleavage sites for this “Trypsin + U” enzyme, which is essentially normal trypsin cleavage + Asp-N cleavage at deamidated glycosite N residues. NGAG search parameters included up to two missed cleavages for “Trypsin + U” digestion; 20 ppm mass tolerance for first search and 6 ppm for main search; carbamidomethylation of C (+57.0215 Da) as a static modification; dynamic modifications of: U→D (−53.9373 Da, representing nascent D residues created by PNGase F-mediated deamidation of glycosite N residues), U not at N termini→N (−54.9213 Da, representing potential glycosites that were not glycosylated and thus not deamidated by PNGase F or cleaved by Asp-N), guanidination of K (+42.0218 Da), aniline at protein C termini, D and E not at N termini, and K and R at any C terminus (+75.0473 Da), two anilines on D and E at protein C termini (+150.0946 Da), guanidination and aniline on K at any C termini (+117.0691 Da) and oxidation of M (+15.9949 Da); six modifications were allowed per peptide, a minimum peptide length of 7 amino acids, and a 1% FDR versus a reversed decoy proteome database. Glycosites in the search data were identified by U at peptide N-termini that had been converted to D.

### 2.8. Intact Glycopeptide Data Analysis

Intact glycopeptide LC-MS/MS data were analyzed by the GPQuest software package developed in-house [43]. Briefly, glycopeptide MS/MS spectra were extracted by assaying the five most abundant MS2 ions for oxonium ions of HexNAc (204.087 Da) and at least one other glycan marker mass (138.055 Da for a fragment of HexNAc, 163.061 Da for Hex, 168.066 Da for HexNAc—2 H_2_O, 274.093 Da for Neu5Ac—H_2_O, 292.103 Da for Neu5Ac and 366.140 Da for HexHexNAc) with 50 ppm tolerance. These spectra were matched with a 10 ppm mass error to the candidate database comprised of glycans identified by PNGase F release and MALDI analysis and glycosite-containing peptides identified by the combined glycosite analyses.

## 3. Results

### 3.1. Quantitative Analysis of CHO N-Glycosites Using Three Glycoproteomic Methods Revealed Complementarity of the NGAG Method

To determine how NGAG fits into the existing framework of glycosite enrichment methodologies we compared it with two widely used glycopeptide enrichment approaches, SPEG and HILIC (Figure 1A), each utilizing different mechanisms to enrich glycosites (Figure 1B–D. We performed these three approaches in parallel with matching starting quantities of the same secreted protein samples from CHO cell cultures. Separately, cell proteomes were also examined by SPEG and HILIC workflows. After protein extraction, trypsinization, and desalting, we used the resulting tryptic peptides from each condition for the various enrichment methods. After each enrichment workflow, we analyzed each set of samples by triplicate single-shot LC-MS/MS runs.

When examining the overlap between the three methods (Figure 2), we first compared the established SPEG and HILIC methods and found that 70.1% of glycosites identified by SPEG were also identified by HILIC whereas 44.2% of those identified by HILIC also were identified by SPEG; furthermore, 37.2% of the 567 total glycosites identified by these two approaches were identified by both methods. Interestingly, the glycosites identified by NGAG exhibited a lower degree of similarity to either of these existing methods, either alone or combined. Specifically, NGAG identified only 126 glycosites in common with SPEG, which was 41.9% of the total SPEG glycosites while NGAG and HILIC overlapped with 36.3% of the HILIC-identified glycosites and only 29.5% of the NGAG identifications. Of the 953 glycosites identified by any of the three methods, only 99 or 10.4% were identified by all three methods. In total, NGAG identified 586 unique sites (Appendix A), HILIC 477 (Appendix A), and SPEG 301 (Appendix A). We attribute, at least in part, the increased performance of NGAG to the more sensitive Q-Exactive mass spectrometer used for this workflow, versus the Velos Pro used for SPEG and HILIC analyses. The 23% increase in the number of glycosites identified for NGAG versus HILIC is only slightly higher than the 12.9% [48] and 10.8% [49] increases in peptide identifications reported previously for mammalian samples with Q-Exactive versus the Velos Pro, while the 94% increase in NGAG over SPEG can more likely be attributed to the extraction methods. The notable lack of overlap of glycosites indicates that NGAG can extract a portion of the glycoproteome not commonly studied up to this point and therefore represents a valuable complementary enrichment methodology for full experimental coverage of the glycoproteome.

Using precursor ion intensity-based label free quantification, we were able to quantitatively compare the relative abundance of glycosite-containing peptides between conditions and enrichment strategies (Figure 3A). Interestingly, NGAG detected 117 glycosites in the 1,3,4-*O*-Bu_3_ManNAc treated samples (548 versus 431). As this phenomenon was not observed in SPEG or HILIC, nor in intact glycopeptide analyses described below, and because many of these glycosites were observed in untreated control samples from SPEG and HILIC, this is most likely due to the stochasticity of single-shot LC-MS/MS and instrument under-sampling.

Overall, Pearson correlation coefficients comparing different samples of the same enrichment method ranged from 0.91 to 0.96, which was higher than comparisons between different enrichment methods applied to the same biological sample. HILIC and NGAG correlate with SPEG to a similar level—NGAG vs. SPEG Pearson coefficients range from 0.50 to 0.58, and HILIC vs. SPEG Pearson coefficients range from 0.52 to 0.55—while NGAG vs. HILIC comparisons exhibit lower Pearson coefficients ranging from 0.32 to 0.36. (Figure 3B). Thus, the biases in glycopeptide capture efficacy between these methods can outweigh biological sources of abundance differences, which is not unexpected but warrants caution, particularly when comparing between glycosite abundances.

Separately, we also performed the SPEG and HILIC workflows on protein harvested from CHO cell lysates, which allowed comparison of glycosylation patterns within the cellular proteome and secretome. Overall, the number of unique glycosites identified from cellular proteins was much higher for both enrichment methods, with 729 cellular vs. 301 secreted in SPEG (Appendix A) and 721 cellular vs. 477 secreted in HILIC (Appendix A) and the cellular vs. secreted glycosites differed in identity and/or abundance in many cases. Parallel analysis of the cellular glycoproteome by SPEG and HILIC workflows revealed that the glycosylation patterns of individual proteins differed between secretome and cellular proteome. Specifically, eight glycosites exhibited glycosylation only in the cell proteome and 26 exhibited glycosylation in the secretome (Figure 3C). These 34 glycosites were distributed across 29 proteins and were identified by both SPEG and HILIC analyses. Most of these proteins exhibited other glycosites with glycosylation in both cellular and secreted samples. This analysis was limited to samples not treated with 1,3,4-*O*-Bu_3_ManNAc.

### 3.2. NGAG Intact Glycopeptide Analysis Revealed Increased Numbers of Secreted Glycosites with Multiple Sialic Acids after Treatment with the Sugar Analog 1,3,4-O-Bu_3_ManNAc

While NGAG presented a new approach to enriching glycosites when it was introduced, its primary purpose is for improved intact glycopeptide analysis by creating a library from these glycosite data, as well as a searchable *N*-glycan library from glycomics analysis of the same samples. We first generated a glycosite-only peptide database from the NGAG glycosite results described above. We next analyzed glycans released by PNGase F treatment of the glycopeptides captured in the NGAG workflow by MALDI-TOF MS. Searching the data from these traces by GlycoPeak finder and matching them to known glycan structures identified 60 unique glycans (Appendix A). These 60 glycans were then compiled into the glycan library, and along with the glycosite library compiled from the NGAG glycosite workflow were then used in searches by a software suite previously developed in house, GPQUEST. Narrowing down the entire proteome and the entire glycome to peptide and glycan databases in this way reduces search space and allows more accurate peptide + attached glycan PSM assignment in less time.

We sought to test this intact glycoproteome workflow by examining glycan-at-glycosite heterogeneity in the CHO cell secretome and how priming sialylation machinery may alter this endpoint by comparing the secretomes of CHO cells with and without 1,3,4-*O*-Bu_3_ManNAc treatment [44,50]. 1,3,4-*O*-Bu_3_ManNAc is an N-acetylmannosamine analog that efficiently crosses cellular membranes and has its ester-linked butyrate groups removed from the core ManNAc by non-specific intracellular esterases [51]. The ManNAc moiety enters the sialylation pathway increasing the glycosylation of cellular [29] and secreted proteins [28] while the butyrate improves recombinant protein production [27]. Notably, these effects are accomplished with negligible effects on the viability of treated cells [27].

We purified intact glycopeptides from duplicate samples of the secretome of untreated control and 1,3,4-*O*-Bu_3_ManNAc treated CHO cells by HILIC without PNGase F treatment, then analyzed them via LC-MS/MS. Searching these data with GPQuest using the NGAG glycosite and glycan libraries resulted in the identification of 686 unique intact glycopeptides (Appendix A), which were comprised of combinations of 55 unique glycans at 193 unique glycosites from 124 different proteins.

A comparison of intact glycopeptide data from 1,3,4-*O*-Bu_3_ManNAc-treated and control CHO cells revealed that overall glycopeptide numbers were not significantly altered by 1,3,4-*O*-Bu_3_ManNAc treatment. The glycan composition of the identified glycopeptides, however, was significantly altered by increased sialyation (Figure 4A). Increased sialylation was evident in two ways: a statistically significant increase in the abundance of glycopeptides with two sialic acids was observed concomitant with a significant decrease in non-sialylated glycopeptides. The abundance of glycopeptides with three and four sialic acids also increased upon 1,3,4-*O*-Bu_3_ManNAc treatment and glycopeptides with one sialic residue correspondingly decreased but these changes were not statistically significant by unpaired *t*-test. There was, however, a statistically significant decrease in a combined group of glycopeptides with zero and one sialic acids, and a significant increase when glycopeptides with two or more sialic acids were grouped. This zero-sum relationship suggested 1,3,4-*O*-Bu_3_ManNAc did not increase overall glycosylation significantly but did increase the likelihood that a glycan would be sialylated. This is further demonstrated by grouping glycopeptides by their attached glycan moieties and calculating the intensity ratio for treated versus control samples (Figure 4B). Almost all glycopeptides with glycans containing zero sialic residues decreased in intensity, with a similar but slightly weaker trend for glycans with one sialic acid residue, while almost all containing two or more residues increased.

Of importance to recombinant protein production, the primary utility of CHO cell culture, two of the three reported glycosites on rhEPO, Asn24 and Asn83, were included in our secretome data—rhEPO was not purified before analysis. These data showed similar sialylation patterns to the overall secretome (Figure 4C), with both glycosites exhibiting increased sialylation after 1,3,4-*O*-Bu_3_ManNAc treatment. Asn83, for example, exhibited a relative decrease in two glycan structures with one sialic acid and relative increases of various structures with 2, 3 and 4 sialic acids.

## 4. Discussion

The largely non-overlapping *glycosite* coverage across the secretome versus SPEG and HILIC in combination with its capacity for *intact glycopeptide* analysis proves NGAG as a valuable glycoproteomics workflow. Overall, we identified a total of 953 unique glycosites from all three of the methods combined with 61% from NGAG analysis, 50% from HILIC, and 32% from SPEG. Strikingly, only 200 of the total 586 (34%) identifications from NGAG were also observed in the other workflows, leaving almost two thirds—or 386 which was more than the entire SPEG dataset—unidentified by either of the two widely used current methods. Meanwhile, SPEG and HILIC overlapped more strongly with each other, with 70% of SPEG glycosites identified by HILIC and 44% vice versa. NGAG identified 586 total glycosites (a 94% increase over SPEG and a 23% increase over HILIC) including 386 not seen in the other workflows. As previously noted, however, we cannot fully attribute this increase in glycosite identifications to NGAG extraction because we used a more sensitive mass spectrometer for analysis in this workflow. The Thermo Scientific Q-Exactive instrument we used for NGAG glycosite and intact glycopeptide analyses is well understood to exhibit increased sensitivity, resolution, and speed versus the Velos Pro instrument we used for SPEG and HILIC glycosite analyses. Shortly after the Q-Exactive was released, Michalski et al. [48] compared these two instruments on triplicate 5 µg injections of HeLa cell digest and reported an increase in peptide identifications of 12.9% and in protein identifications of 27.7% for the Q-Exactive. A later comparison between the two instruments across a range of sample loading amounts was performed by Sun et al. [49] and the newer instrument produced a 10.8% increase in peptide identifications and a 10.1% increase in protein identifications with 1 µg injections of RAW 264.7 cell digest though notably increased improvement over the Velos Pro with lower sample loading. Thus, we must attribute some—but likely not all—of the increase in deglycosylated peptide identifications from the NGAG protocol (94% increase versus SPEG and 23% versus HILIC) to the Q-Exactive instrument.

This large portion glycosites uniquely identified by NGAG suggests that one or more steps of the workflow is biased in a different way than SPEG and HILIC. We believe the observed bias is best attributed to the initial immobilization step of covalent binding between peptide amino termini and aldehyde resin during initial immobilization, as opposed to immobilization via cis carboxyls (SPEG) or polar nature of glycans (HILIC). Indeed, in theory, virtually all tryptic peptides across the proteome possess α-amino groups and should bind aldehyde groups on the solid-phase resin. Highly efficient and robust chemical and enzymatic modifications by guanidine, aniline, PNGase F, and Asp-N combine to produce a reliable, and thorough workflow that is complementary to other approaches. These efficient and robust methods are also amenable to comprehensive analyses of the glycoproteome, such as that employed by our group recently in a larger-scale analysis of the CHO cell glycoproteome using HILIC based glycopeptide enrichment [39]. In this study, our group enriched intact glycopeptides via mixed anion exchange (MAX) cartridges, pre-fractionated them using basic reverse-phase LC (bRPLC) and analyzed them across 24 LC-MS/MS runs for extensive glycoproteome coverage. Using a similar approach to this study, the previous study also utilized a glycosite database from the same samples by a parallel workflow incorporating PNGase F cleavage after bRPLC prior to LC-MS/MS analysis. By this approach, the previous study identified 10,338 intact glycopeptides, comprised of 1162 glycosites across 530 proteins (for comparison, the present study identified 686 unique intact glycopeptides across 193 glycosites on 124 proteins). While it is difficult to compare the two studies due to their difference in scale, the orthogonality we observed between NGAG and HILIC suggests the combined use of these complementary workflows could provide a significant boost to the number of glycoprotein identifications. Alternatively, the use of NGAG instead of HILIC may provide identification of a different subset of the glycoproteome. The complementarity between large scale analyses by different extraction methods may prove particularly useful as we approach complete coverage of the glycoproteome.

The use of multiple proteolytic enzymes to increase proteome coverage has been reported previously [52,53,54] and a recent study showed that dual proteolytic treatment with Trypsin followed by Asp-N increased protein IDs in a yeast proteomics experiment by roughly 10% [55]. While the NGAG protocol does utilize both enzymes, the steps between these proteolysis treatments include blocking of endogenous Asp residues from Asp-N cleavage by treatment with aniline. This is partially reversed by the creation of new Asp residues by PNGase F deglycosylation and deamidation, which are then cleaved by Asp-N for release from solid phase support and analysis. The proteome contains many more Asp than Asn residues [56], let alone Asn residues exhibiting *N*-glycosylation. Thus, while Asp-N cleavage after trypsinization may increase coverage and present a small potential advantage to the NGAG protocol, we expect it does not reach the 10% increase reported previously.

The examination of cellular and secreted samples by the SPEG and HILIC methods revealed dozens of glycosites only observed in the cellular or secreted proteomes. These data are consistent with the idea that glycosites play a role in the secretion of proteins—and that intra- vs. extracellular protein localization depends on glycosylation. Glycosylation affecting secretion has been reported numerous times, including for hyaluronidase 1 [57], *Rhizopus chinensis* lipase [58], and secretion was induced for hydrophobic cutinase and llama antibody fragments after the introduction of glycosites into their genetic codes [59].

The increase in sialylation observed throughout the secretome—and perhaps most importantly, recombinant EPO—supports other literature [27,28,60] in suggesting 1,3,4-*O*-Bu_3_ManNAc to be a viable glycoengineering tool for improved glycosylation patterns in therapeutic protein production. A 350 uM concentration of this analog in growth media increases intracellular sialylation content by almost double that of a 20 mM concentration of ManNAc itself, and is particularly effective in conjunction with overexpression of GnTIV/GnTV/ST6 in CHO cells which increases sialylation of rhEPO by 75% [27]. It should be noted that three *N*-glycosites on EPO—Asn24, Asn38, and Asn83—have been reported previously [61], while we only detected two, omitting Asn38. We believe this is mainly because our analysis was performed without purifying rhEPO. Thus, the overall amount and concentration of rhEPO were lower compared to other studies examining this protein, and the background matrix much more complex.

The identification by NGAG of over 100 glycosites in 1,3,4-*O*-Bu_3_ManNAc treated samples that were not detected in untreated samples was an unexpected result, as the analog primarily affects sialylation and effects on overall glycosylation have not been reported. Because this increase was not observed in the other glycoproteome workflows we employed in this study, and because many of these glycosites were observed in untreated samples by these other methods, we believe it to have been largely stochastic in nature and/or due to under-sampling in the MS instrument. Because we employed single shot non-prefractionated LC-MS/MS runs, label-free non-multiplexed quantification, and because our endpoint analytes were peptides (not proteins) our data are subject to dropouts. In single-shot LC-MS/MS analyses the sample is too complex for the MS instrument to select and sequence all potential peptide precursors—the instrument essentially has too many peptides to sequence and selects somewhat randomly, causing run-to-run variation. This is somewhat suppressed in analyses of pre-fractionated samples as in the previously published large-scale CHO glycoproteome analysis by our group, because the sample complexity after fractionation is reduced, and thus less prone to random ion selection in the instrument. Label-free non-multiplexed quantification/comparison leads to dropouts because each sample is analyzed in a separate LC-MS/MS run, thus allowing run-to-run variation to occur. This has been a key driver in the development of label-based quantification such as stable isotope labeling by amino acids in cell culture (SILAC [62]) and isobaric peptide labels such as tandem mass tags (TMT [63]), in which samples are multiplexed and analyzed in the same LC-MS/MS run. These labeling strategies negate the sample-to-sample variation introduced by precursor ion selection, and while TMT has been shown to reduce identifications in glycoproteomics applications, strategies to overcome this have been reported [64]. Finally, because our endpoint analytes were peptides (or glycopeptides), we lose the grouping and averaging of peptide abundances that occurs in experiments that focus on more conventional protein quantification, where the absence of a peptide from the group that corresponds to a protein is less influential on the final data. Altogether, these points constrained our comparisons to the differences in peptide (deglycosylated peptides in glycosite analysis and intact glycopeptides in that analysis) abundances we observed, rather than presence versus absence. In our intact glycopeptide analysis (Figure 4), we only compared glycopeptides found in both 1,3,4-*O*-Bu_3_ManNAc treated and control. The secretome versus cellular proteome comparison (Figure 3C) is a notable exception where we did focus on presence versus absence, but only for deglycosylated peptides that exhibited the same pattern in both SPEG and HILIC data.

In conclusion, we believe that the NGAG glycosite, glycan, and intact glycopeptide analysis protocol provides a viable, complementary option for glycoproteome analysis. The notable lack of overlap in glycosite identifications it exhibited with the widely used protocols to which we compared it in this study suggests it has the potential to detect a subset of the glycoproteome not previously observed, thus expanding our overall coverage of the glycoproteome in many systems of interest. Its intact glycopeptide analysis approach, meanwhile, enables the further enhancement of measurement of glycan-at-glycosite heterogeneity, which is of critical importance moving forward.

## Figures and Tables

**Figure 1 bioengineering-07-00144-f001:**
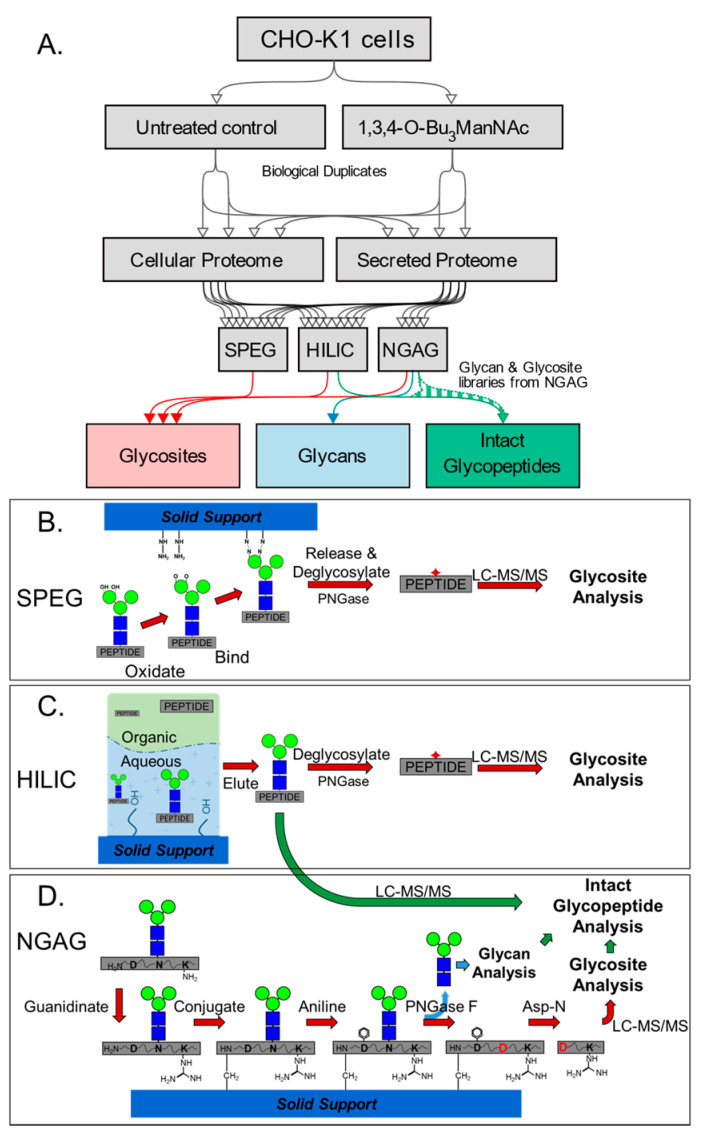
Experimental workflow and schematic of each glycoproteome enrichment method. (**A**) Duplicate samples of Chinese hamster ovary (CHO-KI) cells expressing recombinant human erythropoietin were untreated control or treated with 1,3,4-*O*-Bu_3_ManNAc. Proteins were harvested from the media and digested by trypsin, with equal amounts of peptides used for each glycoproteome workflow. (**B**) In solid-phase extraction of N-linked glycoproteins (SPEG) glycosite identification analysis, glycopeptides were captured by covalent binding of oxidized glycan moieties to hydrazide beads followed by release by PNGase F. (**C**) In hydrophilic interaction liquid chromatography (HILIC) glycosite analysis, glycopeptides were sequestered in the aqueous phase retained by the polar moieties of the column, eluted, then deglycosylated by PNGase F. (**D**) In N-linked glycans and glycosite-containing peptides (NGAG) glycopeptide analysis, glycopeptides were extracted by a stepwise chemoenzymatic process via capture by the peptide N-terminus, deglycosylated by PNGase F, and ultimately released by Asp-N cleavage at the nascent aspartic acid residues for glycosite analysis. *N*-Glycans released by PNGase F were analyzed by MALDI-TOF MS and the resulting glycosite and glycan libraries were combined with intact glycopeptide enrichment via HILIC without PNGase F treatment for efficacious intact glycopeptide identification.

**Figure 2 bioengineering-07-00144-f002:**
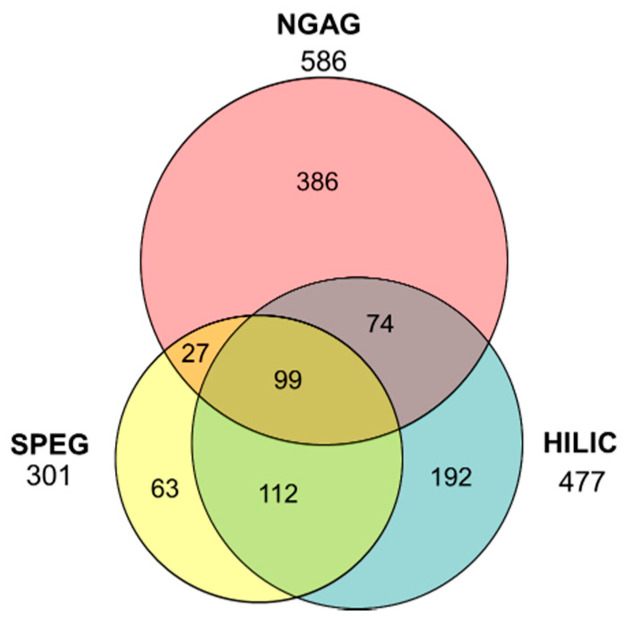
*N*-Glycosites identified from the CHO secretome. SPEG and HILIC showed greater overlap in identified glycosites than NGAG, which exhibited considerable orthogonality. Overall, NGAG identified the largest number of glycosites (586) from the CHO secreted proteome with HILIC identifying 477 and SPEG 301.

**Figure 3 bioengineering-07-00144-f003:**
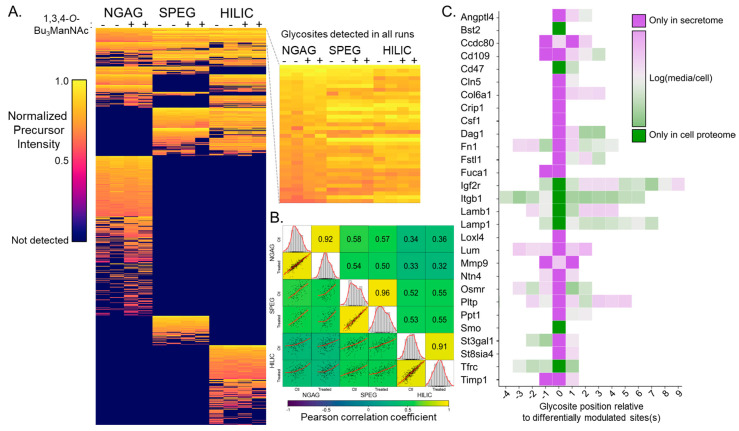
Secretome glycosite quantification by precursor intensity. (**A**) Heatmap showing normalized precursor intensities of all 953 glycosite-containing peptides from each biological replicate from untreated control (−) and 1,3,4-*O*-Bu_3_ManNAc treated (+) samples identified by NGAG, SPEG, and HILIC. Dark blue coloring indicates a glycosite that was not detected in a particular replicate. Glycosites are ordered by the degree of overlap between the three approaches and from high precursor intensity to low. Inset depicts the 35 glycosites that were detected in all replicates to demonstrate the level of agreement between enrichment workflows. (**B**) Correlation matrix of both treated and control conditions, with replicates averaged, for NGAG, SPEG, and HILIC workflows. The top right portion contains the Pearson correlation coefficients for the intersecting conditions, the middle diagonal depicts the distribution of each condition, and the bottom left portion contains scatterplots for the intersecting conditions with a fitted line. Overall, higher correlation was observed between the results of the same workflow from different treatment samples, with notably lower correlation between NGAG and HILIC results. (**C**) Heatmap showing the Log(media/cell) values for 29 proteins (along the *y*-axis) that contained 34 glycosites observed only in cell proteome or only in secretome samples by both SPEG and HILIC analyses. The *x*-axis shows the position of each glycosite within the protein, relative to those observed only in cell proteome or the secretome which was assigned a position of ‘zero’. For several proteins that contained two glycosites in only the cell or secreted proteome, these multiple sites are centered around a zero position. The values are Log ratios of the mean taken from SPEG and HILIC precursor intensities, with glycosites observed in only cell or only secretome assigned nominal values for display.

**Figure 4 bioengineering-07-00144-f004:**
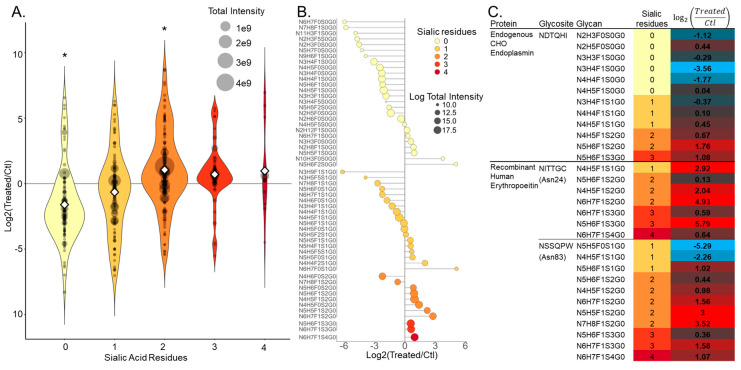
Intact glycopeptide analysis allows comparison of glycan composition caused by treatment with the sugar analog 1,3,4-*O*-Bu_3_ManNAc. Overall, this analog caused a shift toward increased sialylation within identified glycopeptides, including secreted recombinant human erythropoietin (rhEPO). (**A**) Combination violin and dot plot depicting the distribution of Log2 treated/control ratios (*y*-axis) for all detected intact glycopeptides, grouped by the number of sialic acid residues in their glycan moieties (0–4, *x*-axis). The log-transformed intensity ratio for each intact glycopeptide is depicted by a gray dot scaled to the total intensity for that glycopeptide and the mean intensity ratio for each number of sialic acid residues is depicted by a white diamond. The distribution and mean intensity of glycopeptides containing zero or one sialic acid residue decreased in samples treated with 1,3,4-*O*-Bu_3_ManNAc and those containing two to four residues increased. The decrease for glycopeptides containing zero sialic acid residues and the increase those containing two sialic acid residues was statistically significant as measured by an un-paired t-test (statistical significance * *p* < 0.05, un-paired *t*-test). (**B**) Lollipop chart depicting all detected glycopeptides, grouped by glycan moiety (*y*-axis). For each shared glycan, glycopeptide intensities were summed, and the Log2 treated/control ratios calculated (*x*-axis). The color of each dot depicts the number of sialic acid residues within the glycan and the size of the dot depicts their overall intensity. (**C**) A table showing glycosites on two representative proteins bearing out the overall trend. The NDTQHI glycosite on endogenous endoplasmin and the NITTGC (Asn24) and NSSQPW (Asn83) glycosites on rhEPO (Ectopic Human Erythropoietin) exhibiting intensity ratio decreases in glycans with 0 or 1 sialic residue and increases in those with 2–4 sialic residues.

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
