# Peer review of "Comparison of Three Glycoproteomic Methods for the Analysis of the Secretome of CHO Cells Treated with 1,3,4-O-Bu3ManNAc"

_bioengineering, 2020, doi:10.3390/bioengineering7040144_

Round 1
Reviewer 1 Report
The present manuscript by Mertz et al focuses on the NGAG glycoproteomic workflow in comparison to existing glycopeptide enrichment methodologies (HILIC, SPEG) based on a CHO-K1 cell model and optional But3ManNAc treatment. Although the direct comparison would be a valuable contribution for comprehensive glycoproteome analyses and data are of considerable impact, the experimental design, thread and manuscript needs major revision.
The authors and colleagues already published the NGAG method, the story around Bu3ManNAc – also focused on rhEPO sialylation – and the glycoproteomic analysis of the CHO-K1 cells – also distinguished for cell lysate and media protein – elsewhere (e.g. Ref. 20, 29, 32, 37). In fact, only the comparison of enrichment methodologies contributes to the novelty of the manuscript. In this regard, it is not a straightforward design that no data for the CHO cell lysate are shown just for NGAG (the method which is in the focus of this work). Moreover, the measurement with different instruments (deglycosylated peptides from SPEG and HILIC on a Velos Pro, deglycosylated NGAG peptides on a Q-Extactive) are a major bottleneck for a direct comparison of methodologies. Figures and tables are poorly described and lack significant information about the samples and data shown. The discussion section is absolutely underestimated – with almost half portion of result description and no single reference to existing literature. A final / overall conclusion is missing completely. The story around But3ManNAc does not provide a substantial increase in the value of the present manuscript and should be embedded/discussed more thoroughly accomplishing a continuous thread throughout the manuscript.
Please find a list of more details and comments in the attachment.

Author Response
REVIEWER 1
Experimental Design
- It is not straight forward and scientifically inconsistent to measure samples which were directly compared to each other with different mass spectrometers: ‘Deglycosylated peptides from SPEG and HILIC were analyzed on a Velos Pro and deglycosylated peptides from NGAG were measured on a Q-Exactive’
– This is a very valid point. The increase in identifications by NGAG over HILIC (23%) is within the range of 28% increases reported by Sun et al. in 2013. We have edited the manuscript to emphasize the complementary roles of the SPEG, HILIC, and NGAG approaches, and to address the fact that different glycosite identifications and coverage could be due to the use of different mass spectrometers as well as the under sampling issues of LC-MS/MS. Relevant edits can be seen at: p1/line18, p3/L110, p7/L254, p11&12/L390
- Cell lysates were only generated for SPEG and HILIC analyses. Since NGAG is in the focus of this paper, why this is not performed for NGAG? In this regard, Figure 1A implies that cell lysates were analyzed by NGAG as well; which is obviously not correct. – The manuscript has been edited to reflect our primary focus on the secretome, including in the title. The cell lysates were analyzed separately and some interesting findings regarding glycosites found in only the cell proteome vs the secretome presented in Figure 3C. If this is too distracting from the main story, this can be removed. Edits to this point can be seen at: p1/L2,17, p3/L105 & 117, p4/L130,137,139,148,151 p6/L236, Figure 1, p8-9/L292
- An additional Asp-N digestion step was described in the ‘Methods’-section, but not in ‘Introduction’ part for NGAG procedure which begs the question of whether it is really part of the method? According to the authors, this further Asp-N step is required to remove ‘non-modified aspartate residues’ suggesting that aniline reaction with carboxyl residues is not complete?! This leads to the problem that several glycosites were not detectable in the NGAG procedure, since non-modified aspartates can be present e.g. between the immobilized N-terminus and first glycosite or between first and second glycosite on one peptide etc. – We expect the aniline reaction to have been complete, but the first Asp-N step was used as a safeguard against any possible contamination of non-modified Asp residues in downstream steps. We would prefer to have false-negative rather than false-positive results in this regard. This has been explained further p4/L157 and we hope it is suitable.
- The data show no reproducibility or robustness of the applied methods. How big are the deviations within the same approach? Table 1 seems to show biological duplicates. Between both replicates, differences up to 30 identified glycosites were measured invalidating the expression on p.7/58 “586 topping 567”. – The heatmap in Fig. 3A and correlation matrix in Fig. 3B partially address the reproducibility of the applied methods. The heatmap demonstrates the similarity between biological duplicates analyzed by the same method and the correlation matrix demonstrates the correlation between different conditions (after averaging biological duplicates) analyzed by the same method. The legend of Figure 3 has been expanded to explain this (p9/L306).
Scientific Writing
- Figure/table legends:
- Legend of figures and tables should be placed directly above/under the respective item – otherwise its confusing; legends are more a description of results than a description of the data shown there – I believe this occurred during conversion from the .doc to .pdf format in the MDPI system. The legends were placed as you described. We will try to ensure this formatting passes through conversion.
- Figure 3A: I wonder why there a 2 x 2 lanes per enrichment strategy – it is not explained in the figure legend: are these the biological duplicates? – These lanes are indeed biological duplicates, the figure legend has been edited to explain this (p9/L305).
- Figure 3 B: please describe figure in more detail, especially the diagonal pictograms – This is a correlation matrix with Pearson correlation coefficients for intersecting conditions in the top right portion, a histogram of each condition in the diagonal, and scatterplots with fitting lines for the intersecting conditions in the bottoms left portion. The legend has been expanded to describe it in more detail (p9/310)
- Figure 3 C: are these data for one enrichment method only (SPEG or HILIC) or is this a combination? – These data are averaged values from both SPEG and HILIC, including for the glycosites observed in only cell proteome or only secretome samples. The legend has been expanded to describe it in more detail (p9/315)
- Discussion:
- A large part of the discussion is description of data and results; discussion itself is minimal (3 points with 1-5 sentences maximum) – no single literature reference is cited in the discussion (e.g. ‘intra- vs extracellular location plays role in their glycosylation’ – reference to literature data would be helpful and required) – The discussion has been modified and expanded to discuss several new points and now cites several publications. (p11-13)
- Figure 3A: differences in -/+ But3ManNAc lanes can be observed, especially for NGAG – can you comment on this – This is indeed the case, 117 more glycosites were observed in the treated samples, but as this was not observed in SPEG, HILIC, or intact analyses, we believe this to have been an artifact due to stochasticity and/or under sampling in the MS instrument. This has been noted in an edit in section 3.1 p7/L266 and in the discussion. Although speculative and beyond the scope of this report, we believe increased sialylation could improve the efficiency of one or more steps in the NGAG workflow, for example under sampling in the MS instrument. Treatment with 1,3,4-O-Bu3ManNAc thereby allows identification of glycosites we know to exist from their observed presence in the SPEG or HILIC workflows
- You and your colleagues published similar analysis already in 2018 for same cell / protein model with a different approach (Ref. 32) - please discuss your new data in this context – A brief discussion of this previous study and how the present report relate has been included on p12/L408
- Conclusion is missing completely. Do the authors propose the use of one of the three methods or NGAG? – The following conclusion has been added at the end of the discussion, p13:
“In conclusion, we believe that the NGAG glycosite, glycan, and intact glycopeptide analysis protocol to provide a viable, complementary option for glycoproteome analysis. The notable lack of overlap in glycosite identifications it exhibited with the widely used protocols to which we compared it in this study suggests it has the potential to expand overall coverage of the glycoproteome by enabling detection of glycosites not previously observed. Its intact glycopeptide analysis approach, meanwhile, enables further enhancement of measurement of glycan-at-glycosite heterogeneity, which is of critical use moving forward.”
- Please number the supplementary files with S1-S6 accordingly, S5 and S6 are not traceable – The supplementary files have been renamed and their references in the text corrected (p7/L263)
Content
- Introduction: Description of genetic engineering could be reduced (not relevant for the paper) and description of metabolic engineering should be extended regarding e.g. other analogs – We included only one sentence describing genetic glycoengineering approaches, and felt it is of interest to the readership.
- Methods: Manufacturer information are partially missing, e.g. for 1,3,4-O-Bu3ManNAc, medium for CHO cells, etc. –1,3,4-O-Bu3ManNAc was made in house, and the manuscript edited to reflect this p3/L119, and manufacturers for other reagents added throughout p3-5
- Methods: Was EPO purified from the supernatant or just present in the supernatant? Please clearly state in this document the protein sample preparation for supernatants and lysates. – rhEPO was just present in the secretome. The analyses were performed directly on the secretome, not on purified rhEPO. This has been noted in the results and discussion sections (p10/L362, p12/L434), as it may pertain to the fact that we only observed two glycosites on EPO, while a number of studies have reported 3 sites. Previous experiments reported in the literature have seemingly all been performed on purified rhEPO, which improves sensitivity for the targeted protein.
- Which protein concentration did you use in your experiments? Or did you use a defined cell count? – An equal amount of tryptic peptides from each sample (600 µg) was used for each workflow. This has been edited on p3/L114.
- Usage of PNGase F: Used units, manufacturer and incubation time? – We used 1000 units PNGase F from New England Biolabs overnight for each cleavage step, which has been edited at p3/L121, p4/L134,149.
- Methods: Point out which kind of analysis method followed for the respective generated samples. – This has been included for SPEG at p3/L123. It was already included for HILIC at p4/L132,133. It was included for glycan analysis from NGAG, but glycosite analysis was included for this workflow at P4/L152. The LC-MS/MS analysis subsection of Methods has been edited to improve clarity regarding this at p4-5.
- Redundant description of methods in four parts of the paper: introduction, methods, results (see section 3.1) and figure 1: Could be reduced. . – The description has been reduced in section 3.1 p6/L244.
- Expression “one other glycan” (p.5/ lane 207) is not correct – please specify glycan marker masses / oxonium ions or change expression (e.g. [HexNAc – 2 H2O]+ for 168.066 Da) – This has been clarified and expanded p6/L234.
- Figure 1: Glycan analysis for HILIC and NGAG is missing in the figure. – Glycan analysis was only performed for the NGAG workflow. Figure 1 has been edited to reflect this and for other points.• Three N- and one O-glycosylation sites are described for EPO in literature. Why weren’t all three N-glycosites identified? Did one of the three methods identify O-glycopeptides? Would be interesting for the reader. Please comment on that; table 1, column ‘number of oxionium ion containing spectra to be O’. Do you mean O-glycans? – As noted in a response above (bullet point 3 in ‘Content’), it is most likely that we only identified two of the three EPO glycosites (Asn38 and Asn 83, but not Asn 24) because we did not enrich EPO, which was done in previous reports of EPO glycosites. This has been noted in the discussion p13/L435.
- Impact of 1,3,4-O-Bu3ManNAc on viability of CHO cells is not described (reference?). –This analog has a negligible impact on CHO cell viability is as described previously in publications from the Yarema group, and this has been included in an edit at p2/L70
- Global analysis: for what? No results are shown, or discussion was done for this part. – Mentions of global analysis have been removed as the results from this are not particularly relevant, edits at p5/L201, figure 1 p7,
- “with high amounts of overlap between conditions” (p. 7/line 264) is not shown in Suppl. F1, as stated – This passage in the text has been removed, but identified glycans from each sample were similar and identical in some cases. The database constructed from identified glycans has been included as Supplementary table S5.
- Measurement of sialylated N-Glycans in MALDI: a mass difference of sialylated glycans referring to aniline modification can be observed – assuming these are NGAG data (which is not clearly stated in the figure legend); but what about HILIC N-glycans? was there no need of derivatization, how did you assure sia-detection (commonly permethylation or desialylation is necessary!) – As noted above, glycan analysis was not performed on HILIC samples. The incorrect mention of this has been corrected in the methods p3/L125, p4/L135
- Figure S1: Title of axes are missing; are data normalized to 100%? Which condition is ‘the representative spectra” – this is not clearly stated; is there any cut-off for peak identification or what about unlabeled peaks (e.g. at m/z around 1180)? – The data were normalized to 100% for the most intense peak and unlabeled peaks did not produce results in the GlycoPeakFinder search, and likely represent contaminants, artifacts, or adducts that were not captured by the search. The figure has been edited to address your comment and the legend has been expanded.
- Table 1: One lane is missing (Treated 2a) and table description: are these the biological duplicates? – These are biological duplicates, but Table 1 has been removed as it was unclear.
Style
- Missing or additional spaces (prior references, between number and unit, double spaces, e.g. p.9/314-315) – These have been corrected throughout the manuscript.
- ‚N‘ and ‚O‘ of N- and O-glycans should be in italics – This has been corrected throughout the manuscript.
- ‚1,3,4-O-Bu3ManNAc‘-notation should be uniform – This has been corrected throughout the manuscript.
- ‚PNGase F‘-notation should be uniform. – This has been corrected throughout the manuscript.
- Methods: ‚LC/MS/MS analysis‘ and ‚Global and glycosite data analysis‘ sections should be present in two or maybe more paragraphs for lucidity – The LC-MS/MS analysis section has been broken into more paragraphs and rearranged for lucidity, p4&5

Reviewer 2 Report
The manuscript by Mertz and co-workers is about detection and quantitation of glycopeptides in cells / cell culture supernatant. At closer scrutiny it divides in to three only loosely connected parts.
The first one compares two more classical means of glycopeptide enrichment (periodate oxidation-hydrazine beads; HILIC; vs. the self-developed sophisticated NGAG method). The latter does far better than the first two and I want to belief that the authors gave all methods a fair chance by applying the optimally. Then, the interesting or frustrating result is that these methods exhibit a warningly low overlap. However, the results have been obtained with different proteolytic regimes. Trypsin only vs. trypsin + Asp-N. Could that explain the advantage of NGAG?
Given the fact that all three methods rely on the conversion of Asn to Asp I would very much like to read the word de-amidation somewhere. (I recall the case of finding NG as glycosylation site despite some clever precautions https://doi.org/10.1016/j.cell.2010.04.012).
In principle, this part of the manuscript is of interest. If the bioengineering readership is the most interested selection of scientist is yet another question.
The second topic is the comparison of glycosites found in the secretome vs. whole cell lysate. This result is a bit hidden in Figure 3 and the description in the text as well as the legend leaves me behind with question marks. The axis labels are hardly readable and the sentence in line 289 is difficult to understand. In general, I doubt that glycoproteomics can retrieve meaningful results for the comparison of cells vs. supernatant.
Notably, this part only applies the “old” enrichment methods.
The third part in turn relies on intact glycopeptide analysis after HILIC enrichment to compare the effect of a sialic acid precursor on the glycome of an EPO producing cell line. In order to address this question, site-specific glycoproteomics of the entire CHO secretome is certainly the most laborious and expensive approach, which nevertheless only generates nice graphics (Fig. 4 ) and rather complicated table 1 that shows the, hm, what ? – stochastic nature of this approach ? The bottom lines of the table are mysterious. Obviously, the purpose of this exercise was not evaluate the efficacy of 1,3,4-O-Bu3ManNAc for EPO sialylation, as this could have been achieved simply by looking at EPO itself or by analyzing the glycome as such - by whichever method.
There a few further details:
Line 3: reference 3 as the only one here ?
Line 80: simultaneous glycosite + glycan from an intact glycoprotein – sounds very interesting, but only difficult to access reference
L 119: pH ?
L 130; 80 % AcCN … rather low, reason for bad performance ?
L 143; NH3, water …
L 193: many readers may benefit from a brief explanation of the mass of U (which arises from the C-terminal guanidino group)
L 228: LC-MS
L 154: HyperSep is a large family of different products
L 156: This method is of only limited value for sialylated glycans. Is there an actual need to list possible N-glycan compositions in CHO cells? I would assume this is known for years already.
L 250ff: some rather trivial number juggling. Should be reduced to a few relevant findings.
L 264: Expand the “These”
L 276: start of second part without any clear demarcation
L 288f: not understandable (as far is it maybe my fault, I am probably not the only dull person reading this text)
Fig. 3: mixture of 2 unrelated experiments. Furthermore, 3C is barely legible and not readily understandable
Legend: 3B “higher correlation” of which variable? I have a guess, but a clear legend is preferable.
L 320: not significantly altered. Well, the numbers in Table 1, column 3 deviate considerably. I agree that these differences are not significant as they essentially result from the highly stochastic nature of this approach.
157 sialylated glycopeptides found in the one but not the other? Why? Result of different number of sialic acids – hard to believe. Would inspection of MS spectra help to clarify this unsatisfying result?
Fig 4: the term “increased sialic acid residues” is strange.
Overall judgement on section 3.2.: This does not fit to the NGAG part and in may eyes it is an inappropriate approach to answer the question of changes effected by the sialic acid precursor.
The 1,3,4-O-Bu3ManNAc appears in line 98 of the introduction without any reference or commercial source. If it is to be considered seriously, its effect on EPO sialylation should be scrutinized by other means. And then, the data on EPO glycosylation are more interesting than the stochastic hide-and-seek of Table 1.
To conclude: Generally, the paper is well written, a nowadays only rarely encountered pleasure. Another positive aspect: Consecutive line numbering.
I cannot appreciate the stitching together of three different topics – left-over management? Why not narrow down the story to the NGAG comparison?
If the authors are convinced of their sialic acid reagent, it would certainly be worth a separate paper.
Author Response
REVIEWER 2
The manuscript by Mertz and co-workers is about detection and quantitation of glycopeptides in cells / cell culture supernatant. At closer scrutiny it divides in to three only loosely connected parts.
The first one compares two more classical means of glycopeptide enrichment (periodate oxidation-hydrazine beads; HILIC; vs. the self-developed sophisticated NGAG method). The latter does far better than the first two and I want to belief that the authors gave all methods a fair chance by applying the optimally. Then, the interesting or frustrating result is that these methods exhibit a warningly low overlap. However, the results have been obtained with different proteolytic regimes. Trypsin only vs. trypsin + Asp-N. Could that explain the advantage of NGAG? – As noted in response to Reviewer 1, we did use an advanced MS instrument for the analysis of NGAG glycosite samples that could have resulted in the larger number of identifications. We have edited the manuscript to emphasize the orthogonality of the NGAG approach, and to address the fact that increased coverage could be due to the use of a Q-Exactive instrument. Relevant edits can be seen at: p1/line18, p3/L110, p6/L231, p7/L254, p11/L393.
The dual enzyme proteolysis is an interesting question. A report by Dau et al Anal. Chem. Jul 6th 2020 showed that Trypsin + Asp-N did increase protein IDs by roughly 10%. This situation is likely suppressed in the NGAG workflow because of the blocking of endogenous Asp residues and the creation of new Asp residues by PNGase F deglycosylation and deamidation. The proportion of endogenous Asp residues is much higher than even the proportion of Asn residues in the proteome (Brüne et al. BMC Res Notes 2018), let alone Asn residues exhibiting N-glycosylation, thus while Asp-N cleavage after trypsinization may increase coverage, we don’t believe it to be significant. This has been addressed in edits to the discussion on p12.
Given the fact that all three methods rely on the conversion of Asn to Asp I would very much like to read the word de-amidation somewhere. (I recall the case of finding NG as glycosylation site despite some clever precautions https://doi.org/10.1016/j.cell.2010.04.012). – Instances of ‘deamination’ have been changed to ‘deamidation’ p3/L98, p5/L209
In principle, this part of the manuscript is of interest. If the bioengineering readership is the most interested selection of scientist is yet another question.
The second topic is the comparison of glycosites found in the secretome vs. whole cell lysate. This result is a bit hidden in Figure 3 and the description in the text as well as the legend leaves me behind with question marks. The axis labels are hardly readable and the sentence in line 289 is difficult to understand. In general, I doubt that glycoproteomics can retrieve meaningful results for the comparison of cells vs. supernatant.
Notably, this part only applies the “old” enrichment methods. - This description has been made clearer in the figure 3 legend, and the figure as well, but it was not a major focus of the manuscript so we didn’t go into much detail with it.
The third part in turn relies on intact glycopeptide analysis after HILIC enrichment to compare the effect of a sialic acid precursor on the glycome of an EPO producing cell line. In order to address this question, site-specific glycoproteomics of the entire CHO secretome is certainly the most laborious and expensive approach, which nevertheless only generates nice graphics (Fig. 4 ) and rather complicated table 1 that shows the, hm, what ? – stochastic nature of this approach ? The bottom lines of the table are mysterious. Obviously, the purpose of this exercise was not evaluate the efficacy of 1,3,4-O-Bu3ManNAc for EPO sialylation, as this could have been achieved simply by looking at EPO itself or by analyzing the glycome as such - by whichever method. – The main goal of the intact glycopeptide portion of the manuscript was to test and showcase the NGAG intact glycopeptide analysis workflow on a clinically/commercially relevant system (CHO-K1 cells) and treatment that we expected to produce changes in the intact glycopeptide makeup (1,3,4-O-Bu3ManNAc). We hope this portion of the manuscript summarizes our results adequately. Table 1 has been removed as it did not improve clarity of the reporting of the results, p11.
There a few further details:
Line 3: reference 3 as the only one here ? – More references have been added at p1/L26 regarding the role of glycosylation in protein function and pathogenesis.
Line 80: simultaneous glycosite + glycan from an intact glycoprotein – sounds very interesting, but only difficult to access reference – Further references have been added at line 80
L 119: pH ? – We performed this step in 0.1% TFA, and the manuscript has been edited to note this.
L 130; 80 % AcCN … rather low, reason for bad performance ? – We followed previously established protocols that have worked well in our hands.
L 143; NH3, water …– This has been corrected.
L 193: many readers may benefit from a brief explanation of the mass of U (which arises from the C-terminal guanidino group) – The U amino residue represents a residue that will not appear in LC-MS/MS data with common modifications or the modifications introduced in our chemoenzymatic treatments. It was introduced into the proteome database to create cleavage sites for the “Trypsin+U” enzyme, which is essentially trypsin + Asp-N at deamidated glycosites, but only appears with “dynamic modifications” converting them to D or N. The manuscript has been edited to include this point on p6.
L 228: LC-MS – This has been corrected.
L 154: HyperSep is a large family of different products – The Mass spectrometry of glycans subsection has been edited, including to specify that we used HyperSep Hypercarb SPE cartridges (Thermo Fisher Scientific).
L 156: This method is of only limited value for sialylated glycans. Is there an actual need to list possible N-glycan compositions in CHO cells? I would assume this is known for years already. – The method used identified 60 glycans in total and 30 of which were sialylated. The glycan database resulting from this analysis has been included as Supplemental table S5. We hope this suitably addresses your concerns.
L 250ff: some rather trivial number juggling. Should be reduced to a few relevant findings. – This has been reduced to the more relevant findings
L 264: Expand the “These” – This has been expanded upon and moved to the Intact glycopeptide analysis subsection p8/Le310-317
L 276: start of second part without any clear demarcation – Because identification and quantification go hand and hand in these experiments, we believe it to be extraneous to separate the two.
L 288f: not understandable (as far is it maybe my fault, I am probably not the only dull person reading this text) – This has been edited for clarity and more specific data reported.
Fig. 3: mixture of 2 unrelated experiments. Furthermore, 3C is barely legible and not readily understandable– We believe that these experiments are related. They were performed in parallel by the same methodology, the CHO cell lysate samples are related to the secretome analysis, and provided an interesting (in our eyes) comparison. It seems Fig 3C may have been corrupted during file conversions and we have included a new version to try to prevent this. The legend has also been expanded, as with other figures to more completely describe what is depicted in the figure.
Legend: 3B “higher correlation” of which variable? I have a guess, but a clear legend is preferable. – The legends of this and all figures have been expanded for more clarity and deeper description of the figures.
L 320: not significantly altered. Well, the numbers in Table 1, column 3 deviate considerably. I agree that these differences are not significant as they essentially result from the highly stochastic nature of this approach.
157 sialylated glycopeptides found in the one but not the other? Why? Result of different number of sialic acids – hard to believe. Would inspection of MS spectra help to clarify this unsatisfying result? –Table one has been removed as it did not add clarity to the reporting of the results (p9). The sialylated glycopeptides found in one condition but not the other was most likely due to stochasticity as you suggest.
Fig 4: the term “increased sialic acid residues” is strange. – Agreed. This has been corrected, p9
Overall judgement on section 3.2.: This does not fit to the NGAG part and in may eyes it is an inappropriate approach to answer the question of changes effected by the sialic acid precursor. – Section 3.2, describing intact glycopeptide analysis, is a critical component of NGAG. The glycosite and glycan identification from NGAG in section 3.1 provided the databases for matching intact glycopeptides to the intact glycopeptide LC-MS/MS data. This subsection has been edited and rearranged to highlight this and make it clearer, p10-11
The 1,3,4-O-Bu3ManNAc appears in line 98 of the introduction without any reference or commercial source. If it is to be considered seriously, its effect on EPO sialylation should be scrutinized by other means. And then, the data on EPO glycosylation are more interesting than the stochastic hide-and-seek of Table 1. – 1,3,4-O-tManNAc was synthesized in house, as now noted in the methods section p10/L107. Its positive effect on EPO sialylation was also previously demonstrated in reference 20 Yin et al. Biotechnol Bioeng 2017. This has been explained further in new edits to this portion of the introduction at p2/L62.
To conclude: Generally, the paper is well written, a nowadays only rarely encountered pleasure. Another positive aspect: Consecutive line numbering.
I cannot appreciate the stitching together of three different topics – left-over management? Why not narrow down the story to the NGAG comparison? – We hope the reframing of the manuscript in the introduction and in subsection 3.2 to highlight that NGAG analysis includes glycosite, glycan, and intact glycopeptide analysis has satisfactorily addressed this comment.
If the authors are convinced of their sialic acid reagent, it would certainly be worth a separate paper. – As mentioned above, this effect of this analog on sialylation, protein production, and cell viability has been reported in previous publications by the Yarema group including:
- Aich, U., C.T. Campbell, N. Elmouelhi, C.A. Weier, S.G. Sampathkumar, S.S. Choi, and K.J. Yarema, Regioisomeric SCFA attachment to hexosamines separates metabolic flux from cytotoxicity and MUC1 suppression. ACS Chem Biol, 2008. 3(4): p. 230-40.
- Almaraz, R.T., Tian, Y., Bhattarcharya, R., Tan, E., Chen, S.-H., Dallas, M.R., Chen, L., Zhang, Z., Zhang, H., Konstantopoulos, K. & Yarema, K.J. Metabolic flux increases glycoprotein sialylation: implications for cell adhesion and cancer metastasis. Mol. Cell. Proteomics 11, M112.017558 (2012), doi: 10.1074/mcp.M112.017558
- Wang, Q., C.Y. Chung, W. Yang, G. Yang, S. Chough, Y. Chen, B. Yin, R. Bhattacharya, Y. Hu, C.T. Saeui, K.J. Yarema, M.J. Betenbaugh, and H. Zhang, Combining Butyrated ManNAc with Glycoengineered CHO Cells Improves EPO Glycan Quality and Production. Biotechnol J, 2019. 14(4): p. e1800186.

Round 2
Reviewer 1 Report
The manuscript was revised and reviewers’ comments have been addressed (largely) thoroughly. Particularly, description of figure legends and article structure/thread have been improved. A major bottleneck of the study stays the usage of different mass spectrometers within this comparative approach; at least the authors pick up this issue in the text now. Data are of interest for a glyco-readership, so manuscript is worth to be published.
However, there are some aspects which were not addressed/edited satisfactorily (please see attachment - red font color).

Reviewer 2 Report
My points are now essentially answered.
In fact the work is clearer now.
I minor point still is the dummy amino acid "U" with a 4 digit accuracy behind the comma. I still do not understand (and it´s not me alone) where this figure derives from.
Just an arbitrary number ?
I appreciate the honest statement about different Orbitraps used. The conclusion obviously is still a bit arbitrary.
Round 3
Reviewer 1 Report
The manuscript was re-revised and comments have been adressed adequately. Particularly, the quality of the discussion section has been further improved.
Please note the following comments. Apart from this, my overall recommendation is to accept the article in the present form.
COMMENTS:
- Include "cell lysate workflow" also in text format in the Figure 1A legend
- Your used peptide input is rather referred to the applied protein amount of 600 µg. It is still not clear how the protein input was methodically determined (e.g. BCA assay..)
- Your S5 Table includes a "Galactose" and a "Hexose" column as well. "Galactose" column only contains ("0" values) - indicating that all galactose values are recorded in the "Hexose" column already. Please check and correct!